# Cerebrospinal fluid proteome maps detect pathogen-specific host response patterns in meningitis

Anahita Bakochi[1], Tirthankar Mohanty[1], Paul Theodor Pyl[2], Carlos Alberto Gueto-Tettay[1], Lars Malmström[1], Adam Linder[1], Johan Malmström[1]*

[1]Department of Clinical Sciences, Division of Infection Medicine, Lund University, Lund, Sweden; [2]Division of Surgery, Oncology, and Pathology, Department of Clinical Sciences, Biomedical Center, Lund University, Lund, Sweden

**Abstract** Meningitis is a potentially life-threatening infection characterized by the inflammation of the leptomeningeal membranes. Many different viral and bacterial pathogens can cause meningitis, with differences in mortality rates, risk of developing neurological sequelae, and treatment options. Here, we constructed a compendium of digital cerebrospinal fluid (CSF) proteome maps to define pathogen-specific host response patterns in meningitis. The results revealed a drastic and pathogen-type specific influx of tissue-, cell-, and plasma proteins in the CSF, where, in particular, a large increase of neutrophil-derived proteins in the CSF correlated with acute bacterial meningitis. Additionally, both acute bacterial and viral meningitis result in marked reduction of brain-enriched proteins. Generation of a multiprotein LASSO regression model resulted in an 18-protein panel of cell- and tissue-associated proteins capable of classifying acute bacterial meningitis and viral meningitis. The same protein panel also enabled classification of tick-borne encephalitis, a subgroup of viral meningitis, with high sensitivity and specificity. The work provides insights into pathogen-specific host response patterns in CSF from different disease etiologies to support future classification of pathogen type based on host response patterns in meningitis.

*For correspondence:
johan.malmstrom@med.lu.se

Competing interests: The authors declare that no competing interests exist.

## Introduction

Meningitis is a common condition with an estimated annual prevalence of 8.7 million cases globally (*Kassebaum et al., 2017*). In the majority of cases, meningitis is caused by viruses, such as enteroviruses, and is associated with low mortality rates (*Chadwick, 2005*). Certain subtypes of viral meningitis (VM), such as tick-borne encephalitis (TBE), are in contrast associated with higher mortality rates and risk of developing neurological sequelae (*Bogovic and Strle, 2015*). Acute bacterial meningitis (ABM) is one of the leading causes of death due to infectious diseases worldwide and is associated with rapid disease progression, increased risk of long-term neurological sequelae in survivors, and high mortality rates (*van de Beek et al., 2004*; *van de Beek et al., 2006*). The different bacterial and viral pathogens are associated with specific virulence mechanisms that impact the molecular phenotype of the host immune response. This information can be used to diagnose patients with meningitis, which routinely involves lumbar punctures to evaluate several parameters in the cerebrospinal fluid (CSF) such as the number of white blood cells (WBCs) as well as glucose and protein concentrations to differentiate between ABM and VM (*Ross et al., 1988*). Unfortunately, these parameters are relatively non-specific yielding inconclusive diagnostic information (*Garty et al., 1997*; *Lindquist et al., 1988*; *Nigrovic et al., 2002*), and it currently remains unknown if different pathogens and pathogen types can evoke detectable differences in host response proteome.

Extracellular body fluids such as blood plasma, saliva, and CSF are deficient in the machinery required for de novo protein synthesis. The protein constituents of body fluids are generated via active protein secretion from surrounding tissues or from passive leakage derived from normal protein turnover from cells and tissues. During healthy conditions, the concentration of individual proteins in body fluids is maintained via a tightly controlled balance between protein secretion and clearance. This balance is altered in severe infectious diseases such as sepsis or meningitis due to the host responses triggered by the invading pathogen (*Malmström et al., 2016*). In meningitis and sepsis, the immune response normally increases cellular and acellular mediators in the CSF and plasma, which leads to a drastic proteome reorganization (*Karlsson et al., 2018*). In the most severe stages of disease, host immune response becomes overwhelming, leading to organ damage and impaired prognosis (*van de Beek et al., 2004*; *van de Beek et al., 2006*).

Improved definitions of the pathogen or pathogen-type specific host response patterns in CSF could provide insights into specific virulence mechanisms, subsequently leading to the development of new diagnostic and prognostic information. However, detection of pathogen-specific host response patterns requires the analysis of large sample cohorts to compare host response patterns from a similar group of pathogens to all other pathogen types. As the protein composition of body fluids is produced elsewhere, mRNA transcript profiling is less suitable for the detection of host responses. Mass spectrometry (MS)-based protein quantification has become the preferred method for multiplexed and quantitative analysis of proteomes (*Aebersold and Mann, 2016*). The prevailing MS strategy relies on data-dependent acquisition (DDA), where the mass spectrometer sequentially selects and fragments trypsin-generated peptide ions to generate informative daughter fragment ion spectra used for database searching. Although such proteomics experiments enable identification of thousands of peptide ions, the strategy is associated with lower quantitative reproducibility, as in complex samples the number of available peptides exceeds the cycling time of the DDA method. In contrast, the recent development of sequential window acquisition of all theoretical fragment ion spectra (SWATH)-MS generates fragment ion spectra of all MS-measurable peptides to produce a digital representation of the analyzed proteome (*Gillet et al., 2012*). In SWATH-MS, proteome maps are generated based on data-independent acquisition (DIA), followed by protein quantification using a priori established assay libraries (*Teleman et al., 2017*). Different assay libraries can be used to iteratively query the same DIA data in an iterative fashion, which is a considerable advantage as it permits a single data collection step for clinical samples (*Guo et al., 2015*) and focused reanalysis in future studies. Importantly, DIA is associated with a high degree of reproducibility, making it possible to merge data sets analyzed separately in time to perform cross-study comparisons. In this way, new opportunities emerge to construct compendiums of proteome maps from physical biobanks to identify protein patterns in body fluids associated with, for example, disease progression and treatment options.

In this study, we developed a compendium of SWATH-MS CSF proteome maps to provide novel insights into central nervous system (CNS) functioning and host response in a cohort of patients with meningitis. We demonstrate how an extendable compendium of proteome maps supports post-acquisition and iterative data analysis using cell- and tissue-derived assay libraries to define discriminatory protein panels associated with ABM or VM. The results revealed how meningitis generates pathogen-specific changes in the CSF proteome. Furthermore, the compendium of CSF proteome maps supported the identification of a protein panel capable of differentiating between ABM, VM, and TBE, a VM subgroup, with high sensitivity and specificity based on the host response patterns in the CSF.

## Results

### Construction of a compendium of SWATH-MS CSF proteome maps from meningitis patients

Detection of pathogen-specific host responses in the CSF proteome requires comparative analysis of patient sample cases of meningitis caused by different pathogens. Here, CSF was collected by lumbar puncture from a cohort of 135 patients admitted to the hospital with the suspicion of meningitis. Following confirmed diagnosis, the patients were broadly subdivided into ABM (n = 35), neuroborreliosis (BM, n = 7), VM (n = 21), suspected ABM (n = 5), suspected VM (n = 16), and inflammation

without infection (n = 2). In this cohort, ABM was caused by 11 different bacterial pathogens, where the most frequent pathogen was *Streptococcus pneumoniae.* VM was caused by seven different viruses with the highest frequency of TBE and enteroviruses (*Figure 1*). Patients with suspected meningitis but with normal WBC count (<5 × 10⁶/L) and with no clinical signs of infection/inflammation were regarded as a control group (n = 49). CSF from each sample was digested, and peptides were analyzed by DDA-MS for the construction of a CSF proteome assay library and DIA-MS to produce a compendium of proteome maps (*Figure 1a*). The CSF assay library was merged with previously established assay libraries from 28 healthy human organs or primary cells to enable the quantification of proteins enriched in relevant tissues such as brain, plasma, and immune cells. The assay

| | Pathogen | N | Gender - n (%) male | Age (years) - Mean |
|---|---|---|---|---|
| ACUTE BACTERIAL MENINGITIS | - | 35 | 16 (47.1) | 58 |
| Community-acquired | Listeria Monocytogenes | 2 | 1 (50) | 79 |
| | Neisseria Meningitidis | 5 | 3 (60) | 25 |
| | Streptococcus Pneumoniae | 17 | 7 (43.8) | 63 |
| | Streptococcus Pyogenes | 2 | 1 (50) | 72 |
| | Pseudomonas Aeruginosa | 1 | 0 (0) | 59 |
| Post-operative | Bacteroides Fragilis | 1 | 1 (100) | 97 |
| | Enterococcus Faecalis | 1 | 0 (0) | 33 |
| | Escherichia Coli | 1 | 1 (100) | 73 |
| | Staphylococcus Aureus | 3 | 1 (33.3) | 68 |
| | Streptococcus Agalactiae | 1 | 1 (100) | 1 |
| | Streptococcus Salivarius | 1 | 0 (0) | 63 |
| VIRAL MENINGITIS | - | 21 | 10 (50) | 45 |
| | Cytomegalovirus | 1 | 0 (0) | 22 |
| | Enterovirus | 5 | 2 (50) | 26 |
| | Herpes Simplex Virus 1 | 2 | 1 (50) | 61 |
| | Herpes Simplex Virus 2 | 4 | 1 (25) | 38 |
| | Herpes Zoster Ophthalmicus | 1 | 1 (100) | 83 |
| | Tick-borne Encephalitis | 5 | 3 (60) | 53 |
| | Varicella Zoster Virus | 3 | 2 (66.7) | 51 |
| NEUROBORRELIOSIS | - | 7 | 3 (42.9) | 63 |
| HEADACHE CONTROLS | - | 49 | 20 (40.8) | 48 |

**Figure 1.** Method flow chart and patient cerebrospinal fluid (CSF) samples. (a) Fifty microliters of CSF from each sample were prepared and analyzed using both shotgun mass spectrometry (MS) and sequential window acquisition of all theoretical fragment ion spectra (SWATH-MS) for the construction of CSF proteome assay library and production of a digital compendium of SWATH-MS proteome maps. These SWATH-MS maps were post-acquisition interrogated with previously established assay library from 28 healthy human organs or primary cells to enable the quantification of proteins enriched in relevant tissues such as brain, plasma, or immune cells. The SWATH-MS maps were further interrogated with the CSF assay library for determining protein profiles correlating with acute bacterial meningitis, viral meningitis, neuroborreliosis and control samples. (b) A summary showing the number of CSF samples in each sample group, as well as the bacterial or viral strains that caused meningitis. The distribution of gender (as percentage in male) and average age for each group is also presented.

library relied on the protein abundance in the analyzed tissues and provides a statistically significant relationship between proteins and tissues, which was integrated into our results to infer the most likely tissue origin of proteins detected in the CSF. The compendium of SWATH-MS files was subsequently interrogated with the merged human assay library followed by interrogation of assay libraries from the most common pathogens causing meningitis to determine protein profiles correlating with pathogen type (*Figure 1b*).

## Changes in the proteome pattern in CSF during meningitis

A total of 771 unique human proteins were confidently detected and quantified in this analysis. The sample group containing the largest number of unique proteins was ABM (n = 719 versus n = 544 in controls), followed by VM (n = 550) and BM (n = 478). On average, the total protein intensities in ABM were 2, 2.7, and 4.3 times higher than in BM, VM, and the control group, respectively (*Figure 2—figure supplement 1*). This reflects an elevated protein concentration in CSF, associated in particular with bacterial meningitis. Analysis of the subcellular compartment of the detected proteins revealed that the majority of proteins are secreted (between 37% and 48% across all samples), followed by membrane-bound proteins, which comprised 18–24% of the total protein content (*Figure 2—figure supplement 2a*). The average intensity for the secreted proteins was 5.9-fold higher in ABM versus control and 2.7–2.9-fold higher in ABM when compared to BM or VM (*Figure 2—figure supplement 2b*). The distribution of quantified proteins in relation to sample groups is presented as a heat map in *Figure 2a*. The sample clustering clearly subdivides the sample cohort into four distinct clusters representing predominantly the ABM, VM, and control samples. The row-wise color coding indicates inferred tissue origin based on the information from the assay library. The most distant sample cluster includes in principle all the ABM samples. The remaining sample clusters are more similar, but the general trend supports subdivision of most of the VM samples from the control samples. We observe that numerous neutrophil-, plasma-, and brain-associated proteins constitute on average 15–20% of the protein intensity, where, in particular, the neutrophil proteins increased during ABM (*Figure 2—figure supplement 3*). Plasma proteins are known to be major constituents of CSF under physiological conditions (*Guldbrandsen et al., 2014*), and presence of brain-associated proteins in CSF is likely related to protein turnover in the brain. Statistical analysis between sample groups reveals 79 statistically induced or repressed proteins (*Figure 2b*–d). The majority of significantly altered proteins were associated with neutrophils (32% in ABM and 14% in VM), the brain (44% in ABM and 32% in VM), and some plasma proteins including several acute-phase proteins (*Figure 2—figure supplement 4a, b*). In addition, several neutrophil-associated proteins were induced ≥64-fold (yellow dots) but not statistically significant, indicating a large degree of variation associated with these proteins. For each group, the average intensities of all neutrophil proteins show a significant increase only in ABM compared to controls (*Figure 2f*), whereas there is a significant decrease in brain-associated proteins in ABM, VM, and TBE (*Figure 2e*). Among the gene ontology (GO) terms enriched for the repressed proteins in ABM was 'regulation of synapse maturation and assembly,' suggesting changes on the cellular level in neuronal circuits during ABM (*Figure 2—figure supplement 4c*). Of the induced proteins, notable GO terms were 'regulation of apoptotic signaling pathway,' 'defense response to bacterium,' and 'glial cell development.' These suggest increased apoptotic pathways and enriched molecular processes specialized in combating bacteria in the CSF of patients with ABM, followed by increased development of glial cells, which are involved in maintaining chemical homeostasis and act as the immune cells of the CNS (*Ransohoff and Brown, 2012*).

## Cross-comparison of CSF protein composition between ABM, BM, and VM

To evaluate differences in the host response between the sample groups, we plotted the differentially expressed proteins in scatter plots, with the corresponding log2 fold change for ABM/control on the x-axis and log2 fold change for BM/control on the y-axis (*Figure 3a*). This cross-plot was repeated for ABM against VM (*Figure 3b*) and VM against BM (*Figure 3c*). Proteins that were statistically significant or associated with high fold changes (≥64) in both sample groups are colored in black, otherwise only in one color (red; ABM, orange; BM and green; VM). This cross-comparison reveals that ABM and VM evoke a more similar response compared to BM. In total, ABM was

**Figure 2.** Cerebrospinal fluid proteome analysis. (a) The quantified proteins (rows) and samples (columns) were clustered and visualized in a heat map. Top horizontal color bar classified sample groups, and left vertical color bar classified the human tissue assignments for each protein. The data was row-wise normalized by Z-score transformation, and column and row clustering was performed with ward.D and canberra clustering criterions. (b–d) Differentially expressed proteins between control and acute bacterial meningitis (ABM) (b), neuroborreliosis (BM) (c), and viral meningitis (VM) (d) are shown in volcano plots. Statistically significant proteins (Hochberg-corrected p-values≤0.05 and log2 fold change ≤ −2 and ≥2) are labeled in red. Statistically non-significant proteins with a high fold change of ≥64 are labeled in yellow. (e, f) Log2 scaled average intensities of brain and neutrophil-associated proteins for ABM, BM, VM, tick-borne encephalitis (TBE), and control are presented. The significance of the changes was calculated with a standard Student's t-test and marked with asterisk (one or two star significance).

The online version of this article includes the following figure supplement(s) for figure 2:

**Figure supplement 1.** Overview of average protein content and intensities.

**Figure supplement 2.** Predicted subcellular compartments of detected proteins.

**Figure supplement 3.** Overview of average protein content and intensities.

**Figure supplement 4.** Analysis of differences in the cerebrospinal fluid (CSF) of patients with meningitis.

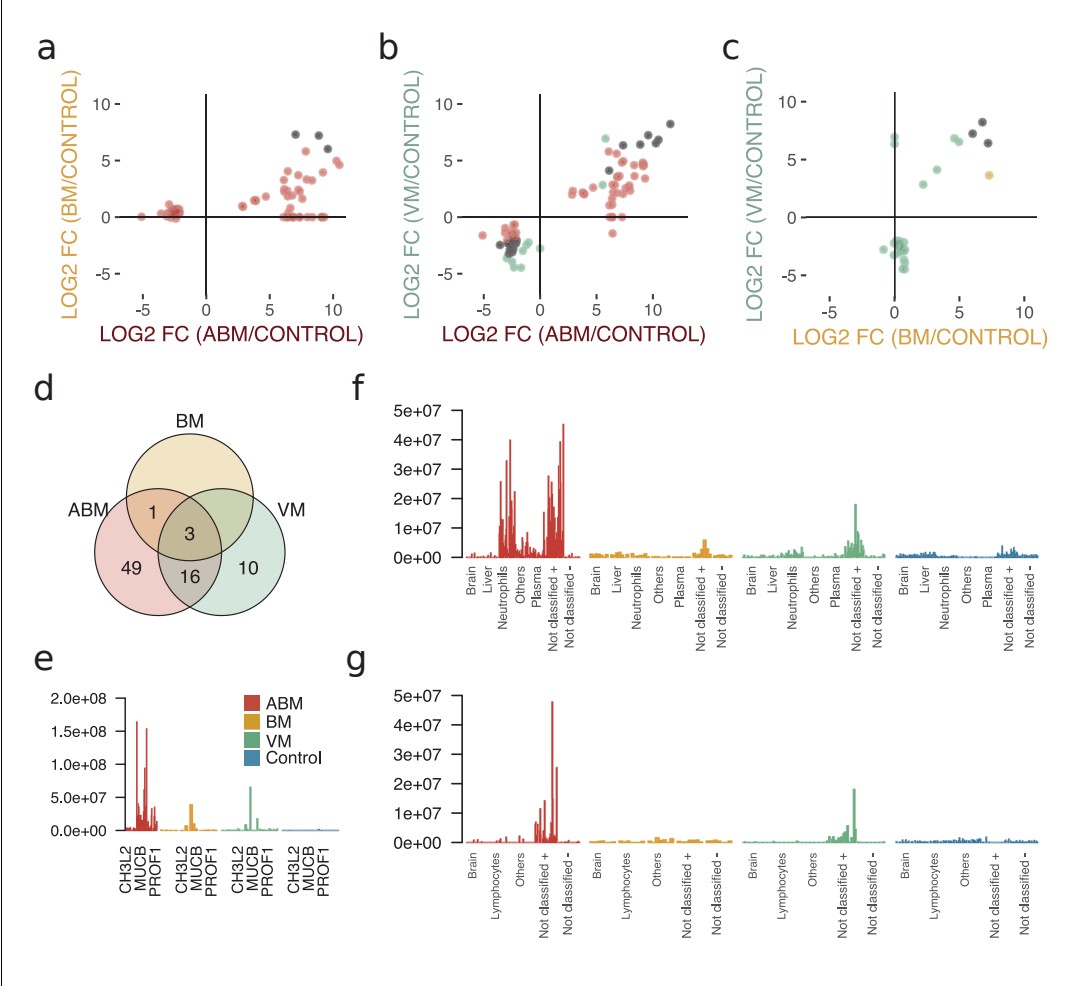

**Figure 3.** Cross-comparison of cerebrospinal fluid (CSF) protein composition between acute bacterial meningitis (ABM), neuroborreliosis (BM), and viral meningitis (VM). (a–c) Significantly regulated proteins (Benjamini–Hochberg-corrected p-value≤0.05 and log2 fold change of ≤ −2 and ≥2) together with proteins with high fold change of ≥64 for all three groups were selected, and their fold changes of two disease group at a time were plotted against each other: ABM versus BM (a), ABM versus VM (b), and BM versus VM (c). The points are colored based on their significance in either one group only (red: ABM; orange: BM; green: VM) or in both (black). Proteins absent in one group were plotted at the null line for that group for visualization purposes. (d) Venn diagram summarizes the overlap of the proteins shown in (a–c) for the different groups. The intensities of the three proteins shared in all three groups (e), 49 proteins specific to ABM (f), and 10 proteins specific to VM (g) are shown as bar plots for every CSF sample. Proteins not classified to a tissue were further divided into two groups depending on if they were downregulated (-) or upregulated (+) compared to control.

associated with 49 unique statistically significant proteins and VM 10 proteins and 16 proteins were shared between the disease groups (*Figure 3d*). Only three proteins, chitinase-3-like protein 2 (CHI3L2), immunoglobulin mu heavy chain disease protein (MUCB), and profilin-1 (PROF1), were found at higher levels in all the three diseased groups compared to control (*Figure 3e*), although the intensities for these proteins are substantially higher in the ABM samples. Previous studies have shown an association between these proteins and neurological disease (*Møllgaard et al., 2016*; *Narayanan et al., 2016*; *Opsahl et al., 2016*). The inferred tissue assignment and average intensities for the 49 ABM-specific proteins in all samples are presented in *Figure 3f* and for the 10 VM-specific proteins in *Figure 3g*. We observed that neutrophil proteins are distinctive for ABM, but not for VM. In addition, ABM also led to increased levels of acute-phase proteins such as serum amyloid a-1 (SAA1) and C-reactive protein (CRP). In contrast, VM resulted in increased concentrations of C4b-binding protein and C-X-C motif chemokine 10 (CXCL10), where the latter chemokine is elevated in plasma during viral infections.

## Predictive proteomic patterns using LASSO regression modeling

To define host response protein patterns specific for pathogen type, we used LASSO regression to select discriminatory protein intensities from the compendium of CSF proteome maps. In total, we included the four broader groups, ABM, BM, VM, and the control group. In addition, we included TBE as this group was associated with a distinct molecular host response. We used fourfold cross-validation to build a model for each of the five groups to distinguish the groups from each other. To assess the stability of our model, we repeated the process 100 times (runs), generating 2000 models in total (4 folds $\times$ 5 groups $\times$ 100 runs). All of the tested groups had an area under the receiver operating characteristic curve (AUROC) above 0.80, and two of them had an AUROC above 0.90 (*Figure 4a*). The protein profiles for ABM and control were reproducible across the 100 runs and specific, with a mean AUROC of 0.96 for ABM and 0.95 for controls. The models for BM and VM had a lower mean AUROC of 0.85 and 0.80, respectively. TBE displayed the most stable and discriminatory AUROC of 0.87 of all specific causative-agent subgroups, indicating that TBE evokes a distinct host response. In total, 18 proteins were detected in ≥90% of the 100 runs in every fold and had a nonzero weight, which were consistently used in the LASSO models for discriminating different sample groups. For these 18 predictive proteins, we visualized the distribution of the respective protein contribution to the models as box plots colored by the average weight (coefficient) over all 100 runs (*Figure 4c*). Proteins with a nonzero coefficient had a positive influence on the model and are regarded as predictive proteins for the disease group as shown by the fill color of the box plots. The average intensity of the 18 proteins shows the differences in the abundance levels in their respective sample group (*Figure 4d*, *Figure 4—figure supplement 1*). 5 of the 18 proteins were in general detected in higher amounts in ABM and BM and include proteins such as gelsolin (GEL), cathepsin D (CATD), and transthyretin (TTHY) that are involved in neutrophil degranulation according to the Reactome Pathway Database. Nine of the proteins were exclusively elevated in ABM and include proteins involved in inflammatory response (A2GL, HPT A1T1), fibrin clot formation (HEP2, PROS), LPS binding (CD14), and regulation of the complement cascade (CFAH). Notably, these proteins are all found in lower concentrations in the control samples, and these proteins were mostly predictive for ABM and/or the control samples. In contrast, BM, VM, and TBE in general have less discriminatory weight coefficient. For TBE, monocyte differentiation antigen CD14 was elevated compared to the controls. Antithrombin-III, a serine protease inhibitor that regulates the blood coagulation cascade, was associated with a high weight coefficient and a relatively high abundance level in the VM group. In conclusion, these results demonstrate that LASSO regression can select a set of predictive proteins to classify different disease etiologies in meningitis, which was not possible by any of single-protein biomarker candidates quantified in this study.

## Longitudinal follow-up of protein levels in ABM and subarachnoidal hemorrhage

Meningitis is a disease that is associated with a risk of developing severe neurological sequelae in survivors, and some studies have shown that the levels of certain proteins remained high in the CSF from non-survivors (*Goonetilleke et al., 2010*), such as chitotriosidase, complement C1q tumor necrosis factor-related protein 9, and haptoglobin. To investigate changes of the CSF proteome over time, we extended the compendium of proteome maps with an additional total of 36 CSF samples from ABM (six patients) and from patients with subarachnoidal hemorrhage (SAH, four patients). From these patients, there were multiple CSF samples collected up to 10–13 days after admittance to hospital (*Figure 5*a). Four LASSO-predicted proteins for ABM remained elevated until the end of the 10-day period (*Figure 5b*), suggesting that sampling over a longer time period is required to observe the protein levels returning to baseline. The brain proteins with significantly lower levels in ABM started to increase during the extended follow-up times peaking around day 5 (*Figure 5c*). In contrast, the proteins with significantly lower abundance in ABM compared to control cases show a slight increase over time (*Figure 5d*), whereas the proteins with higher levels in ABM compared to control decreased in ABM over the 10 day-period (*Figure 5e*). The levels of neutrophil proteins in ABM start to decrease but showed a high degree of variability (*Figure 5f*). In all cases, protein levels were low or absent in SAH. Interestingly, the LASSO-predicted proteins used for discriminating ABM tend to stay elevated even after the patients were released from hospital care. These results indicate an ongoing complex host response process that is unique to infectious neurological disease, such as

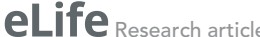

**Figure 4.** Predictive proteomic profiling using a LASSO regression model. (a) Predictive proteomic profiling was performed on all detected and manually curated proteins (n = 771) by LASSO regression model to assign predictive scores to discriminate acute bacterial meningitis (ABM), neuroborreliosis (BM), viral meningitis (VM), tick-borne encephalitis (TBE), and control samples from each other. The samples were split into four randomly selected folds, and the modeling was repeated 100 times. The average area under the curve (AUC) of the sensitivity and specificity of the

*Figure 4 continued on next page*

*Figure 4 continued*

predicted model is presented in a receiver operating characteristic (ROC) curve. (**b**) The variation of the AUC across the 100 repetitions is shown as box plots. (**c**) All proteins that were detected in each fold and in ≥90% of the 100 repetitions (18 proteins) were selected as important proteins for the construction of the LASSO prediction model, and their value (e.g., log-transformed, centered, and scaled protein abundances) plotted as box plots. The fill color of the box plot represents the weight coefficient assigned to each individual protein, where blue and red represent positive or negative effect of that protein for prediction of that specific sample group. (**d**) The selected 18 proteins are presented in a heat map, with sample groups as rows and sample group averages for each protein as columns.

The online version of this article includes the following figure supplement(s) for figure 4:

**Figure supplement 1.** Visualization of the 18 predictive proteins in volcano plots.

ABM, and is absent in other types of neurological trauma such as SAH. Data of protein levels over a longer period of time could possibly help in understanding neurological sequelae that are followed after some cases of meningitis.

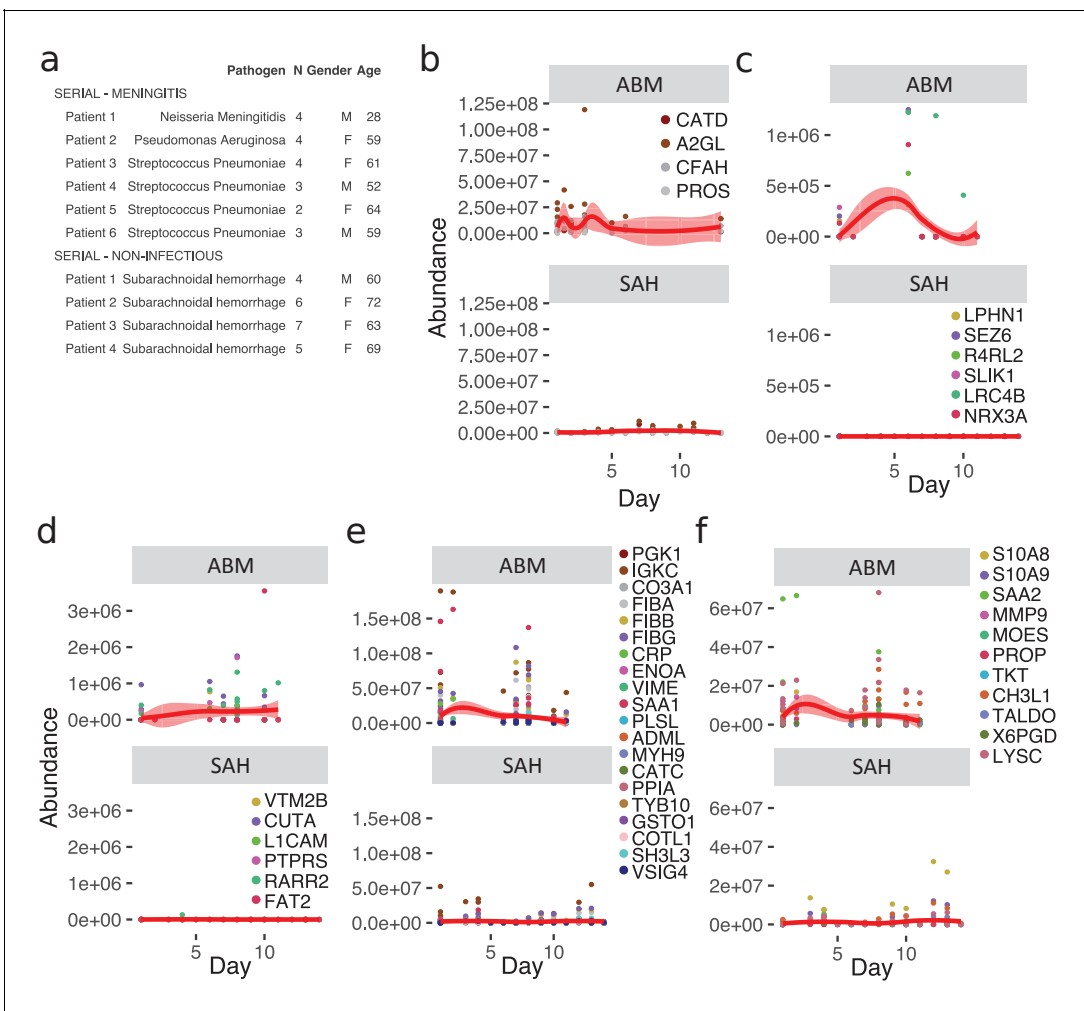

**Figure 5.** Longitudinal investigation of protein abundances in acute bacterial meningitis (ABM) and subarachnoidal hemorrhage (SAH). (**a**) Summary of the cerebrospinal fluid (CSF) samples collected from patients with ABM (n = 6) or SAH (n = 4) for longitudinal analysis of CSF proteome is presented. (**b–f**) Based on previous results, five groups of proteins were selected for longitudinal analysis in ABM up to 10 days and in SAH up to 13 days after admittance to hospital: four proteins selected by LASSO as predictive and indicative of ABM (**b**), brain-associated proteins downregulated in AMB (**c**), downregulated proteins not classified to a tissue and specific to ABM (**d**), upregulated proteins not classified to a tissue and specific to ABM (**e**), and neutrophil proteins upregulated in ABM (**f**).

## Discussion

Here, we present a resource of extendable compendium of CSF proteome maps from a cohort of meningitis patients to define specific host response patterns in meningitis. The quantified proteins have a median coefficient of variation (CV) for technical replicates below 20% as previously shown (*Guo et al., 2015*). The compendium presented in this study comprised in total 171 CSF samples with different sample- and time-dimensions generated from <50 µL of CSF. In SWATH-MS, the post-acquisition targeted data analysis strategy using previously established assay libraries confirms the presence and relative abundance of proteins. Once acquired, the compendium of SWATH-MS maps can be iteratively reanalyzed in silico to test new hypotheses. In this study, we used a highly curated assay library to search for tissue- or cell-enriched proteins informative for host response patterns associated with ABM, BM, TBE, and VM.

In severe infectious diseases, several factors may influence the type and magnitude of the host response, such as type of infecting pathogen, host immune status, and time of infection. The dysregulated host responses during sepsis and meningitis can be detrimental to the host (*Iskander et al., 2013*; *Ward et al., 2008*). At the same time, altered protein composition of CSF provides an opportunity to probe disease status of the complex interplay of factors driving the host response. Damage derived from the infectious process and/or the host response may generate detectable protein changes associated with, for example, organ damage (*Malmström et al., 2015*). In this study, we used LASSO regression to define protein patterns capable of discriminating between ABM, BM, VM, TBE, and controls. Based on an assay library containing information of tissue- and cell-enriched proteins, the different disease groups had noticeable differences in protein patterns associated with neutrophils, blood plasma, and proteins predominantly present in the brain. The increased levels of neutrophil proteins are a consequence of infiltrating activated neutrophils, known to occur predominately during ABM (*Hoffman and Weber, 2009*; *Tunkel and Scheld, 1993*). Neutrophils may release decondensed extracellular DNA coated with antimicrobial proteins called neutrophil extracellular traps (NETs) (*Moorthy et al., 2016*). A recent study showed high levels of NETs in the CSF of patients with pneumococcal meningitis and that disrupting NETs using DNase I significantly reduces bacterial load, demonstrating that NETs contribute to the pathogenesis of pneumococcal meningitis (*Mohanty et al., 2019*). It is noteworthy that both ABM and VM introduce reduced levels of brain-specific proteins, although the overlap of the reduced proteins between ABM and VM is small. The reason for this reduction is not clarified, although it was previously reported that brain proteins decrease in CSF during other neurological disorders, such as Huntington's disease (*Fang et al., 2009*).

The large number of disease-causing pathogens in meningitis leads to high heterogeneity in both patients and disease progression, and this variability can be seen in other severe infectious diseases such as sepsis. In order to account for this clinical diversity in medical research, traditional individual biomarker studies can be replaced with multiprotein panels to provide better coverage of the underlying disease. The LASSO regression model resulted in 18 predictive proteins. Proteins predictive of ABM include CATD and complement factor H (CFAH), both of which are known plasma proteins involved in clearance of specifically bacterial infections (*Bewley et al., 2011*; *Haapasalo et al., 2012*). Leucine-rich alpha-2-glycoprotein (A2GL) has previously been suggested to be an inflammatory marker, and significantly higher levels of A2GL were reported in the CSF of children with ABM compared to febrile controls (*Chong et al., 2018*). Other studies have shown that the CSF levels of alpha-1 antitrypsin (A1AT) and apolipoprotein E (APOE) were higher in ABM compared to controls (*Hoffman et al., 1989*; *Gutteberg et al., 1986*; *Wang et al., 2012*). Furthermore, antithrombin-3 (ANT3), one of the predictive proteins for VM, has previously been shown to have a broad-spectrum antiviral activity for various human cytomegaloviruses and herpes simplex viruses (*Quenelle et al., 2014*). These results together indicate that large-scale multiprotein panels can yield biological and clinical insights that are difficult to achieve with traditional statistical analyses.

The high reproducibility of SWATH-MS is a consequence of the DIA, which generates fragment-ion spectra from all MS-measurable peptides from a proteome. This feature can support extension of the compendium to generate a digital representation of physical biobanks. In total, the sample cohort in this study consists of patients infected with 19 different pathogens, where the three largest groups were *S. pneumoniae*, enteroviruses, and TBE. Among these pathogen groups, the LASSO regression generated the highest AUC for TBE although the sample size was relatively small. There

are currently no biomarkers for detecting and diagnosing TBE (*Bogovic and Strle, 2015*). Future extension of the compendium will enable further investigations of pathogen-specific host responses for other pathogens in CSF. In addition, the opportunity to reanalyze the compendium in silico based on improved assay libraries will provide opportunities to find new protein patterns correlating with other types of clinical information such as disease outcome or the development of neurological sequelae. This goal can be achieved through either a data-driven process where samples are clustered based on similarities in the proteome changes or in a hypothesis-driven fashion where the compendium is interrogated for protein profiles correlating with, for example, pathogen type, infection time, or disease severity. The increasing efforts to construct complete assay libraries of the human proteome (*Kim et al., 2014*; *Matsumoto et al., 2017*; *Zolg et al., 2017*) will provide improved definition of proteins enriched in particular cells and tissues and will enable quantification of additional proteins, post-translational modifications (*Rosenberger et al., 2017*), and proteolytically processed proteins from the already established compendiums of proteome maps. Ultimately such a resource can be anticipated to enable improved correlation between host responses and detectable changes in CSF and potentially blood plasma to identify molecular markers that can be used for the development of new diagnostic-, prognostic-, and treatment-decisive information for severe infectious diseases.

## Materials and methods

### Patients and CSF samples

CSF samples (n = 171) from a total of 139 patients from two different cohorts were analyzed. Patients enrolled in a prospective study at the Clinic for Infectious Diseases, Lund University Hospital, Lund, Sweden, between March 2006 and November 2009 (as previously described; *Linder et al., 2011*), with a clinical suspicion of meningitis who underwent a lumbar puncture and where CSF samples were collected, were included. The patients were categorized into the following groups: ABM (n = 35), BM (n = 7), VM (n = 21, of which TBE n = 5), suspected ABM (n = 5), suspected VM (n = 16), and inflammation with no infection (n = 2). Control patients had a suspected meningitis, but were declared healthy after displaying a normal CSF WBC count ($<5 \times 10^6$/L) (n = 49). In addition, longitudinal samples were collected from six of the previously mentioned ABM patients (6 original samples and additional 14 longitudinal samples) and from four patients with SAH (22 longitudinal samples).

### CSF sample preparation

A constant volume of 50 µL of each CSF sample was used to correlate to the protein concentration present in each sample. Samples were heat-inactivated by incubation on a heat-block for 5 min at 80°C to kill any microorganisms present in the samples, and then transferred into lysing matrix tubes (Nordic Biolabs) containing 90 mg silica beads (0.1 mm). The cells were homogenized with a cell disruptor (BeadBeater, FastPrep 96, MP Biomedicals) twice for 180 s. The samples were incubated for 30 min at 37°C in 10 M urea, 1 M ammonium bicarbonate (ABC), and 1 µg trypsin (sequence grade modified trypsin porcin, Promega) for denaturation and tryptic cleavage. Samples were further incubated in 10 M urea and 1 M ABC for 30 min, after which large unfolded proteins were spun down by centrifugation. The supernatants were reduced by incubation for 60 min at 37°C in 500 mM tris(2-carboxyethyl) phosphine (TCEP, Sigma-Aldrich) and alkylated by incubation with 500 mM 2-Iodoacetamide (IAA, Sigma-Aldrich). Samples were diluted in 100 mM ABC and incubated overnight in 1 µg trypsin, after which trypsin was inactivated by addition of formic acid. C18-columns (Vydac UltraMicro Spin Silica C18 300 Å) were used according to manufacturer's instructions to clean up and concentrate the peptide samples.

### Bacterial strains and sample preparation

For the generation of pathogen assay libraries, bacterial isolates (n = 1 of *Escherichia coli*, *Enterococcus faecalis*, *Streptococcus pyogenes*, *Streptococcus agalactiae*, *Listeria monocytogenes*, *Pseudomonas aeruginosa*, *Staphylococcus aureus*, and n = 4 of *S. pneumoniae*) were taken from frozen stocks and grown overnight at 37°C and 5% $CO_2$ on blood agar plates. Streaks of each plate were grown in Todd–Hewitt broth (30 g/L, Difco Laboratories) supplemented with yeast extract (6 g/

L, Difco Laboratories). The bacterial cells were centrifuged and the pellet was washed three times in wash buffer (50 mM Tris-HCl, 150 mM NaCl, pH 7.6, Medicago). The final pellet was resuspended to a final concentration of $2 \times 10^9$ CFU/mL. The pellet was dissolved in ice-cold LC-grade water and heat-inactivated by incubation at 90°C for 5 min. The homogenization, denaturation, in-solution digestion, reduction, alkylation, and sample clean-up were performed as for the CSF sample preparation.

## LC-MS/MS analysis of CSF samples

All peptide analyses were performed on a Q Exactive Plus mass spectrometer (Thermo Fisher Scientific) connected to an EASY-nLC 1000 ultra-high-performance liquid chromatography system (Thermo Fisher Scientific). For shotgun analysis, peptides were separated on an EASY-Spray column (Thermo Scientific; ID 75 µm × 25 cm, column temperature 45°C). Column equilibration and sample load were performed using constant pressure at 600 bar. Solvent A was used as stationary phase (0.1% formic acid). Solvent B (mobile phase; 0.1% formic acid, 100% acetonitrile) was used to run a linear gradient from 5% to 35% over 60 min at a flow rate of 300 nL/min. One full MS scan (resolution 70,000 @ 200 m/z; mass range 400–1600 m/z) was followed by MS/MS scans (resolution 17,500 @ 200 m/z) of the 15 most abundant ion signals (TOP15). The precursor ions were isolated with 2 m/z isolation width and fragmented using higher-energy collisional-induced dissociation at a normalized collision energy of 30. Charge state screening was enabled and unassigned or singly charged ions were rejected. The dynamic exclusion window was set to 10 s. Only MS precursors that exceeded a threshold of 1.7e4 were allowed to trigger MS/MS scans. The ion accumulation time was set to 100 ms (MS) and 60 ms (MS/MS) using an AGC target setting of 1e6 (MS and MS/MS).

For DIA, peptides were separated using an EASY-spray column (Thermo Fisher Scientific; ID 75 µm × 25 cm, column temperature 45°C). Column equilibration and sample load was performed at 600 bar. Solvent A was used as stationary phase (0.1% formic acid). Solvent B (mobile phase; 0.1% formic acid, 100% acetonitrile) was used to run a linear gradient from 5% to 35% over 120 min at a flow rate of 300 nL/min. A full MS scan (resolution 70,000 @ 200 m/z; mass range from 400 to 1200 m/z) was followed by 32 MS/MS full fragmentation scans (resolution 35,000 @ 200 m/z) using an isolation window of 26 m/z (including 0.5 m/z overlap between the previous and next window). The precursor ions within each isolation window were fragmented using higher-energy collisional-induced dissociation at normalized collision energy of 30. The automatic gain control was set to 1e6 for both MS and MS/MS with ion accumulation times of 100 ms (MS) and 120 ms (MS/MS). The obtained raw files were converted to mzML using MSConvert (*Kessner et al., 2008*). The MS proteomics data have been deposited to the ProteomeXchange Consortium via the PRIDE partner repository with the dataset identifier PXD023174.

## LC-MS/MS analysis of bacterial samples

Peptide analysis was performed as for the CSF samples, with the exception that an Acclaim PepMap 100 pre-column (Thermo Scientific, C18, 3 µm, 100 Å; ID 75 µm × 2 cm) was coupled to the column and the linear gradient of between 5% and 35% acetonitrile in aqueous 0.1% formic acid was run for 120 min instead. The MS proteomics data have been deposited to the ProteomeXchange Consortium via the PRIDE partner repository with the dataset identifier PXD024904.

## CSF shotgun MS analysis

The shotgun MS data was searched with Trans-Proteomic Pipeline (TPP, v4.7 POLAR VORTEX rev 0, Build 201405161127) using X!Tandem against the UniProt human reference proteome (UP000005640, Oct-2015, reviewed and canonical proteins only), and for generation of decoy proteins a reverse approach was used. Cysteine carbamidomethylation was considered as a fixed and methionine oxidation as a variable modification, and enzyme specificity was set for trypsin to allow two missed cleavage sites. Variable acetylation of the n-terminae, S-carbamoylmethyl-cysteine cyclization of the n-terminal cysteines as well as pyro-glutamic acid formation from glutamic acid and glutamine was also allowed by X!Tandem. The precursor mass tolerance thresholds were set to 20 ppm and the fragment mass tolerance to 50 ppm. The raw files were gzipped and Numpressed (*Teleman et al., 2014*) and converted to mzML format using msconvert from ProteoWizard (v3.05930 suite, [*Chambers et al., 2012*]).

## CSF assay library creation and DIA analysis

Assay libraries were created using the Fraggle-Franklin-Tramler workflow (*Teleman et al., 2017*). In brief, fragment spectra from TPP search results were interpreted by the software tool fraggle and assembled into a retention time normalized consensus assay library with a Franklin-derived multilevel false-disovery rate (FDR) of <0.01. Assays were generated by the software tool Tramler and contain only the 3–6 most intense fragments within the mass range of 350–2000 m/z and do not fall within the precursor isolation window (*Deutsch et al., 2012*). The assay library was stored in traML format. For DIA analysis, DIANA v2.0.0 was used (*Teleman et al., 2015*) with a 20 ppm extraction window. The generated data was manually curated to remove various immunoglobulin variable chain proteins.

## Shotgun MS analysis and assay library creation of bacterial samples

A snakemake workflow was created for building generic transition library (GTL) from Thermo DDA experiments. The raw spectral library creation was achieved by a tandem of spectrum clustering, DDA searches and OpenSwath protocols. Initially, a set of DDA raw files were converted to mascot generic format by ThermoRawFileParser software (*Hulstaert et al., 2020*). Secondly, spectra files were clustered through Mass Rarity Clustering approach by MaRaCluster (*The and Käll, 2016*). Ursgal package (*Kremer et al., 2016*) was used as interface for searching the consensus spectra against data's UniProt reference proteome using five search engines, namely MSGFPlus (version 2019.07.03), MS Amanda (version 2.0.0.17442, *Dorfer et al., 2014*), Comet (version 2019.01.rev5, *Eng et al., 2013*; *Eng et al., 2015*), X! Tandem (version alanine, *Craig and Beavis, 2004*), and OMSSA (version 2.1.9, *Geer et al., 2004*). Optional Met Oxidation (UniMod: 35), Asp and Gln deamidations (UniMod: 7), along with the fixed Cys carbamidomethylation (UniMod: 4) modifications, were considered in this study. Individual engine results were validated by percolator (v3.4.0, *Käll et al., 2007*), while the Combine FDR algorithm was implemented for combining results from all search engines (*Jones et al., 2009*). Moreover, a threshold of 1% peptide FDR was set for decisive candidate inclusion. In the final step, GTL database was built by all peptide's b- and y-ions extraction and adding decoy assays, as described previously (*Escher et al., 2012*; *Collins et al., 2013*; *Röst et al., 2014*; *Schubert et al., 2015*). In this implementation, we use eight ion types: b, y, b(2+), y(2+), b-$H_2O$, y-$H_2O$, b-$NH_3$, and y-$NH_3$.

## Human tissue atlas

The human tissue atlas was used to statistically assign all detected proteins (n = 771) to human tissue based on abundance. A total of 12 tissue assignments were used, which included the following tissues and cell types: adipose, brain, liver, nerve, erythrocytes, lymphocytes, macrophages, neutrophils, platelets, and plasma. Proteins associated to other tissues that were available via the atlas are depicted here as 'others.' A protein was depicted as 'not classified' if the abundance could not be statistically associated to only one tissue or if it was missing from the atlas altogether. The tissue assignments of certain proteins used in further analyses in this study were compared and matched against other, publicly available and published protein tissue-assignment repositories (*Supplementary file 1*).

## Statistical analysis

For statistical analyses, a Benjamini–Hochberg-corrected t-test was used. GO annotations were performed by using two unranked lists of proteins with GOrilla (*Eden et al., 2009*). For the LASSO modeling, the proteomic data was first log-transformed, scaled, and centered (i.e., log10 of the abundances, subtraction of the per-protein mean and division by the per-protein standard deviation). This data was then used as the input to the LASSO implementation of the LiblineaR package of the R statistical programming language (R version 3.6.1; LiblineaR version 2.10–8). A fourfold cross-validation approach was used for model building, where the samples were split randomly into four balanced groups (same proportion of pathogen type as the whole cohort). For each of the four groups, the model was trained on 3/4 of the data and the performance was evaluated on the remaining 1/4, and this modeling was performed 100 times. Proteins that received a nonzero weight coefficient (as in the value of that protein has an influence on the model) in ≥90% of all models were selected for further analysis.

## Acknowledgements

We thank Filip Årman for his help with the bioinformatics when generating the pathogen libraries. *Funding*: This work was supported by the Swedish Research Council (project 2015-02481 and project 2018-05795), the Crafoord Foundation (grant 20100892), Stiftelsen Olle Engkvist Byggmästare, the Wallenberg Academy Fellow program KAW (2012.0178 and 2017.0271), European Research Council starting grant (ERC-2012-StG-309831), and the Medical Faculty, Lund University. The computations and data handling were enabled by resources provided by the Swedish National Infrastructure for Computing (SNIC) at SSC and LUNARC partially funded by the Swedish Resource Council through grant agreement no. 2018-05973.

## Additional information

### Funding

| Funder | Grant reference number | Author |
| --- | --- | --- |
| Knut och Alice Wallenbergs Stiftelse | 2017.0271 | Johan Malmström |
| Vetenskapsrådet | 2015-02481 | Johan Malmström |
| Vetenskapsrådet | 2018-05795 | Johan Malmström |
| Vetenskapsrådet | 2018-05973 | Johan Malmström |
| Crafoord Foundation | 20100892 | Johan Malmström |
| Stiftelsen Olle Engkvist Byggmästare | 2012.0178 | Johan Malmström |
| European Research Council | ERC-2012-StG-309831 | Johan Malmström |
| Lund University Medical Faculty Foundation | | Johan Malmström |

The funders had no role in study design, data collection and interpretation, or the decision to submit the work for publication.

### Author contributions

Anahita Bakochi, Conceptualization, Data curation, Formal analysis, Visualization, Methodology, Writing - original draft, Writing - review and editing; Tirthankar Mohanty, Conceptualization, Data curation, Formal analysis, Methodology, Writing - original draft, Writing - review and editing; Paul Theodor Pyl, Data curation, Formal analysis, Writing - original draft, Writing - review and editing; Carlos Alberto Gueto-Tettay, Data curation, Formal analysis, Methodology, Writing - review and editing; Lars Malmström, Conceptualization, Data curation, Supervision, Writing - original draft, Project administration, Writing - review and editing; Adam Linder, Conceptualization, Data curation, Supervision, Funding acquisition, Writing - original draft, Project administration, Writing - review and editing; Johan Malmström, Conceptualization, Supervision, Funding acquisition, Investigation, Writing - original draft, Writing - review and editing

### Author ORCIDs

Anahita Bakochi https://orcid.org/0000-0001-8144-8525
Lars Malmström http://orcid.org/0000-0001-9885-9312
Johan Malmström https://orcid.org/0000-0002-2889-7169

### Ethics

Human subjects: The medical ethics committees (Institutional Review Boards) of the Lund University approved of the study (decision number 790/2005 and 2016/672), and all samples were taken with the informed consent of the participants or next of kin.

Decision letter and Author response
Decision letter https://doi.org/10.7554/eLife.64159.sa1
Author response https://doi.org/10.7554/eLife.64159.sa2

## Additional files

### Supplementary files

• Supplementary file 1. Cross-referencing the tissue assignments based on Malmström et al. (unpublished work). The tissue assignments based on Malmström et al. (unpublished work) for proteins in *Figures 3f, g*, *4c,* and *5b–f* and *Figure 2—figure supplement 4a, b* matched and compared to publicly available and published protein tissue assignment repositories.

• Supplementary file 2. The full data generated from 112 data-independent acquisition MS-runs (acute bacterial meningitis [ABM]: n = 25, neuroborreliosis [BM]: n = 7, viral meningitis [VM]: n = 21 of which tick-borne encephalitis [TBE]: n = 5 and controls: n = 49). The data was manually curated to remove immunoglobulin variable chain proteins.

• Supplementary file 3. The full data generated from the data-independent acquisition MS-runs from the longitudinal study. The cohort consists of longitudinal samples collected from six acute bacterial meningitis (ABM) patients (6 original samples used in this study and additional 14 longitudinal samples) and from four patients with subarachnoidal hemorrhage (SAH, 22 longitudinal samples). The data was manually curated to remove immunoglobulin variable chain proteins.

• Transparent reporting form

### Data availability

The mass spectrometry proteomics data have been deposited to the ProteomeXchange Consortium via the PRIDE [1] partner repository with the dataset identifier PXD023174.

The following datasets were generated:

| Author(s) | Year | Dataset title | Dataset URL | Database and Identifier |
|---|---|---|---|---|
| Bakochi A, Mohanty T, Pyl PT, Gueto-Tettay CA, Malmström L, Linder A, Malmström J | 2021 | Cerebrospinal fluid proteome maps detect pathogen-specific host response patterns in meningitis | https://www.ebi.ac.uk/pride/archive/projects/PXD024904 | PRIDE, PXD024904 |
| Bakochi A, Mohanty T, Pyl PT, Gueto-Tettay CA, Malmström L, Linder A, Malmström J | 2020 | Cerebrospinal fluid proteome maps detect pathogen-specific host response patterns in meningitis | https://www.ebi.ac.uk/pride/archive/projects/PXD023174 | PRIDE, PXD023174 |

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
