## [Decision Letter]

**Acceptance summary:**

The Reviewers and Editors were pleased with the revised manuscript, which has added useful analyses to better appreciate the value of the described MS-based proteomic methodology for characterising a translationally important biological system.

**Decision letter after peer review:**

Thank you for submitting your article "Cerebrospinal fluid proteome maps detect pathogen-specific host response patterns in meningitis" for consideration by *eLife*. Your article has been reviewed by 2 peer reviewers, and the evaluation has been overseen by a Reviewing Editor and Bavesh Kana as the Senior Editor. The following individual involved in review of your submission has agreed to reveal their identity: Martin Beck (Reviewer #2).

Essential Revisions:

1. In some cases the actual pathogens may have left their peptides in CSF. Have the authors tried to detect them by a species wide search? Of course, this may require shotgun data.

2. The authors may want to comment on their proteome coverage in the main text. I understand that this is somewhat delicate for a body fluid, but at least the number of proteins confidently detected would be interesting.

3. It would be helpful in Figure 2 and 3 if the authors could highlight specific proteins in the volcano plots. For instance where are some of the predictive proteins shown in Figure 4c, such as A1AT, APOE, GELS, TTHY, etc?

4. A brief discussion is warranted regarding whether the CSF protein-pathogen associations found in this work have been previously reported in the literature, or represent entirely novel associations.

5. It would be helpful for the authors to perform some type of analysis to depict the localization of the proteins detected (e.g., classically secreted proteins versus membrane versus intracellular). For instance I already see in Figure 4c that ANT3 is mitochondrial. Is there an enrichment of classically secreted proteins in this dataset?*Reviewer #2 (Recommendations for the authors):*

Bakochi et al have developed a swath MS panel to quantify the proteome content of cerebrospinal fluid. They used it to measure a large meningitis patient panel that had been clinically classified into different bacterial and viral pathogens. In particular bacterial infection resulted in an increased protein concentration and neutrophil protein content, while in general brain but i.e. also apoptosis-related proteins were found to be variable. The proteomic profiles of CSF are distinct from a control group and, based on a selected subset of proteins, allow to discriminate the type of pathogen with decent sensitivity and specificity. It is demonstrated to those changes last for a surprisingly long period of time in patients after clinical display.

The manuscript is technically well done, well written and presented. The strength of this study is that it capitalizes on the strength of swath MS in a biological system where protein quantification is without any alternative. Another strength is the relatively large patient panel, although it breaks down to smaller numbers for some of the less frequent pathogens. The panel described here may be used to assess the disease status of patients and maybe further explored for diagnostic potential in the future.

In some cases the actual pathogens may have left their peptides in CSF. Have the authors tried to detect them by a species wide search? Of course, this may require shotgun data.

The authors may want to comment on their proteome coverage in the main text. I understand that this is somewhat delicate for a body fluid, but at least the number of proteins confidently detected would be interesting.

Reviewer #4 (Recommendations for the authors):

Bakochi et al. present a proteomic analysis of cerebrospinal fluid proteins in control patients or in individual sample cases of meningitis caused by different pathogens. Pathogen-specific changes to the CSF proteome was observed. The authors identify specific proteins that could classifying the type of pathogen that caused meningitis. Major strengths of this work include (1) rigorous proteomic analysis, (2) an unusual patient cohort characterized by a diversity of pathogens (bacterial, viral, neuroborreliosis), (3) the ability to perform direct comparisons of distinct pathogen-induced proteomic changes in CSF, and (4) generation of a predictive model based on the proteomics dataset. The data as presented support the conclusion that distinct pathogens induce shared as well as distinct changes to the CSF proteome. No technical weaknesses were noted. However, the likely impact of this work on the field remains limited. There have been many other proteomic studies of the CSF to identify meningitis biomarkers. While this work builds on these previous proteomic datasets, it remains unclear whether this dataset is more robust, reproducible, or representative versus some of the other published datasets (PMID 26040285, 28765563, 20608875, etc.; reviewed in 31404562).

1. It would be helpful in Figure 2 and 3 if the authors could highlight specific proteins in the volcano plots. For instance where are some of the predictive proteins shown in Figure 4c, such as A1AT, APOE, GELS, TTHY, etc?

2. A brief discussion is warranted regarding whether the CSF protein-pathogen associations found in this work have been previously reported in the literature, or represent entirely novel associations.

3. It would be helpful for the authors to perform some type of analysis to depict the localization of the proteins detected (e.g., classically secreted proteins versus membrane versus intracellular). For instance I already see in Figure 4c that ANT3 is mitochondrial. Is there an enrichment of classically secreted proteins in this dataset?

---

## [Author Response]

Essential Revisions:1. In some cases the actual pathogens may have left their peptides in CSF. Have the authors tried to detect them by a species wide search? Of course, this may require shotgun data.

We completely agree. In fact, at the onset of this project our main aims were to detect both pathogen-specific and -induced host patterns in the CSF as well as peptides from the infecting pathogen directly in these CSF patient samples to increase predictive power of our analysis. To accomplish this goal, we generated 8 different pathogen assay libraries for the most prevalent pathogens; *Escherichia coli, Enterococcus faecalis, Streptococcus pyogenes, Streptococcus agalactiae, Listeria monocytogenes, Pseudomonas aeruginosa, Staphylococcus aureus* and *Streptococcus pneumoniae*. The pathogens were by grown in vitro and analyzed by DDA and DIA to generate the libraries. We have now made the source data and the assay libraries publicly available and deposited to the ProteomeXchange Consortium via the PRIDE partner repository with the dataset identifier PXD024904 as a resource for the community. Description of the bacterial sample preparations and data analysis for the library generation is in the "Materials and methods" section. Unfortunately, the samples in this cohort were sterile filtered, which is a possible explanation why we could not detect the proteomes from the microorganisms in these samples.

2. The authors may want to comment on their proteome coverage in the main text. I understand that this is somewhat delicate for a body fluid, but at least the number of proteins confidently detected would be interesting.

As requested, have added the numbers of confidently detected and quantified proteins in total, as well as for each sample group separately (ABM, BM, VM and controls). This information can be found in the Results section "Changes in the proteome pattern in CSF during meningitis" on rows 129-131.

3. It would be helpful in Figure 2 and 3 if the authors could highlight specific proteins in the volcano plots. For instance where are some of the predictive proteins shown in Figure 4c, such as A1AT, APOE, GELS, TTHY, etc?

The volcano plots in Figure 2 are relatively small, adding individual protein labels would make the figure unreadable and visually disagreeable. Instead, we added a new figure supplement "Figure 4—figure supplement 1" (Visualization of the 18 predictive proteins in volcano plots), where we have reproduced the same volcano plots from Figure 2 in a larger format. In these volcano plots we have included all the protein labels for each of the 18 predictive proteins as requested. We have further mentioned the new "Figure 4—figure supplement 1" in the manuscript in the Results section "Predictive proteomic patterns using LASSO regression modeling" on row number 215.

As for the scatter plots in Figure 4, none of the predictive proteins were found within the limits of the scatter plots, and therefore no alterations have been made to these figures.

4. A brief discussion is warranted regarding whether the CSF protein-pathogen associations found in this work have been previously reported in the literature, or represent entirely novel associations.

We have searched the literature for CSF protein-pathogen associations of the LASSO-generated predictive proteins. For several of these proteins we were unable to find any cross-references that would demonstrate an association between the CSF levels of one of our predictive proteins and meningitis. These proteins include ANT3, CATD, CD14, CFAH, CLUS, DKK3, ENPP2, FCGBP, GELS, HEP2, HPT, ITIH2, PROS, SCG1 and TTHY. For the other predictive proteins (A1AT, A2GL and APOE) we found references that suggest elevated protein levels are found in the CSF during meningitis. This matter is discussed in the Discussion section, on row numbers 295-300.

5. It would be helpful for the authors to perform some type of analysis to depict the localization of the proteins detected (e.g., classically secreted proteins versus membrane versus intracellular). For instance I already see in Figure 4c that ANT3 is mitochondrial. Is there an enrichment of classically secreted proteins in this dataset?

We have now done an analysis of the predicted subcellular compartment of the detected proteins for each sample group (ABM, BM, VM and control). This analysis is presented as grouped bar plots in a new "Figure 2—figure supplement 2". In "Figure 2—figure supplement 2A" we have reported the percentage of proteins belonging to each compartment per sample group, and in "Figure 2—figure supplement 2B" the total average intensity of each proteins belonging to corresponding subcellular compartment. This analysis is now also discussed in the manuscript in the Results section "Changes in the proteome pattern in CSF during meningitis" on rows 134-139.